# Ventral pallidal encoding of reward-seeking behavior depends on the underlying associative structure

Jocelyn M Richard[1]*, Nakura Stout[1], Deanna Acs[1], Patricia H Janak[1,2,3]*

[1]Department of Psychological and Brain Sciences, Krieger School of Arts and Sciences, Johns Hopkins University, Baltimore, United States; [2]Solomon H Snyder Department of Neuroscience, Johns Hopkins School of Medicine, Johns Hopkins University, Baltimore, United States; [3]Kavli Neuroscience Discovery Institute, Johns Hopkins University, Baltimore, United States

**Abstract** Despite its being historically conceptualized as a motor expression site, emerging evidence suggests the ventral pallidum (VP) plays a more active role in integrating information to generate motivation. Here, we investigated whether rat VP cue responses would encode and contribute similarly to the vigor of reward-seeking behaviors trained under Pavlovian versus instrumental contingencies, when these behavioral responses consist of superficially similar locomotor response patterns but may reflect distinct underlying decision-making processes. We find that cue-elicited activity in many VP neurons predicts the latency of instrumental reward seeking, but not of Pavlovian response latency. Further, disruption of VP signaling increases the latency of instrumental but not Pavlovian reward seeking. This suggests that VP encoding of and contributions to response vigor are specific to the ability of incentive cues to invigorate reward-seeking behaviors upon which reward delivery is contingent.

DOI: https://doi.org/10.7554/eLife.33107.001

*For correspondence:
richardj@umn.edu (JMR);
patricia.janak@jhu.edu (PHJ)

Competing interests: The authors declare that no competing interests exist.

## Introduction

The tendency to seek rewards, like many adaptive behaviors, is influenced by multiple dissociable decision-making processes. These decision-making processes have different costs and benefits and may be differentially vulnerable to perturbations that contribute to psychopathology. The ventral pallidum (VP) is a critical node in the neural circuitry underlying reward-related behaviors (*Creed et al., 2016*; *Smith et al., 2009*), including relapse to drug and alcohol seeking (*Farrell et al., 2018*; *Kalivas and Volkow, 2005*; *Saunders et al., 2015*). Neurons in the VP are known to respond to a variety of reward-related stimuli, including primary rewards (*Itoga et al., 2016*; *Tindell et al., 2006*), Pavlovian cues predicting reward delivery (*Smith et al., 2011*; *Tindell et al., 2005*; *2009*), cues predicting reward availability (*Richard et al., 2016*), and cues indicating specific appropriate reward-seeking actions (*Ito and Doya, 2009*; *Tachibana and Hikosaka, 2012*). Further, activity in VP is critical for normal levels of cue-elicited reward seeking across a variety of paradigms (*Leung and Balleine, 2015*; *Mahler et al., 2014*; *Prasad and McNally, 2016*).

Yet, there is little consensus on the primary role of VP in driving reward-seeking behaviors, and how VP signaling contributes to reward seeking across behavioral domains and rewarding outcomes. While historically the VP has been suggested to act as a motor expression site for a 'limbic-motor' interface (*Heimer et al., 1982*; *Mogenson et al., 1980*) due to its close connections with motor output regions, more recent work has suggested a greater role for VP in reward processing itself. For instance, VP responses to cues have been argued to contribute to reward seeking by representing expected reward value (*Tachibana and Hikosaka, 2012*), but have also been suggested to encode

**eLife digest** Sounds or other cues associated with receiving a reward can have a powerful effect on an individual's behavior or emotions. For example, the sound of an ice cream truck might cause salivation and motivate an individual to stand in a long line. Cues may prompt specific actions necessary to receive a reward, for example, approaching the ice cream truck and paying to get an ice cream. This is called instrumental conditioning. Some cues predict reward delivery, without requiring a specific action. This is called Pavlovian conditioning. Pavlovian cues can still prompt actions, such as approaching the truck, even though the action is not required. But exactly what happens in the brain to generate these actions during the two types of learning, is unclear.

Learning more about these reward-driven brain mechanisms might help scientists to develop better treatments for people with addiction or other conditions that involve compulsive reward-seeking behavior. Currently, scientists do not know enough about how the brain triggers this kind of behavior or how these processes lead to relapse in individuals who have been abstinent. Basic studies on the brain mechanisms that trigger reward-seeking behavior are needed.

Now, Richard et al. show that a greater activity in neurons, or brain cells, in a part of the brain called the ventral pallidum predicts a faster response to a reward cue. In the experiments, some rats were trained to approach a certain location when they heard a particular sound in order to receive sugar water, a form of instrumental conditioning. Another group of rats underwent Pavlovian training and learned to expect sugar water every time they heard sound even if they did nothing. Both groups learned to approach the sugar water location when they heard the cue, despite the different training requirements.

Richard et al. measured the activity of neurons in the ventral pallidum when the rats in the two groups heard the reward-associated sound. The experiments showed that the amount of activity in the brain cells in this area predicted whether a rat would approach the sugar-water delivery area and how quickly they would approach the reward after hearing the cue. The predictions were most reliable for rats that had to do something to get the sugar water. When Richard et al. reduced the activity in these cells they found the rats took longer to approach the reward source, but only when this action was required to receive sugar water. The experiments show that the ventral pallidum may provide the motivation to undertake reward-seeking behavior.

DOI: https://doi.org/10.7554/eLife.33107.002

state and action values (*Ito and Doya, 2009*). Previously, we showed that VP neurons encode the incentive value of cues, as defined by their ability to invigorate instrumental reward-seeking actions (*Richard et al., 2016*).

Here, we examined VP encoding of cue-driven reward seeking in two different behavioral models, in which cues generate superficially similar behavioral patterns based on similar levels of and variations in reward expectancy, after training with distinct underlying reward contingencies. The first model consists of Pavlovian conditioning, in which an auditory tone predicts the delivery of sucrose to a reward port. While delivery of the reward is not contingent on the animal's behavior during the cue, over conditioning the animal learns to approach the reward port during the cue, prior to reward delivery. The second model consists of a discriminative stimulus task, similar to that reported previously (*Richard et al., 2016*), but modified to generate superficially similar behavior to that following Pavlovian conditioning. In this task, the animal is presented with an auditory cue (the discriminative stimulus, DS), which indicates the availability of sucrose reward from the port *if* the animal enters the reward port during the cue period. We found that although VP neurons respond robustly to reward-related cues in both tasks, these cue responses are only robustly predictive of reward seeking vigor when reward delivery is contingent upon that behavior. This suggests that VP neuron responses to cues contribute to reward seeking, not by driving locomotor vigor more generally, but by encoding an underlying decision process that is reflected in the latency of instrumental but not Pavlovian responses.

## Results

### Ventral pallidum population encoding of discriminative stimuli is more robust than Pavlovian cues

To assess the degree to which VP encoding of cue-elicited reward seeking depends on the underlying task structure, we trained rats in either an instrumental task (the 'DS task') or in Pavlovian conditioning. In the DS task, entry into the reward port during the DS (an auditory cue lasting up to 10 s) resulted in delivery of liquid sucrose (10%), whereas port entries during an alternative auditory cue (the non-reward stimulus [NS]) had no programmed consequences. During Pavlovian conditioning, presentations of one auditory cue (the CS+) predicted delivery of liquid sucrose at the end of the cue, irrespective of the animal's port entry behavior, whereas presentations of an alternative auditory cue (the CS–) did not predict sucrose delivery. In both tasks, rats learned to enter the reward delivery port more quickly (*Figure 1A–B* and *Figure 1—figure supplement 1A–B*) and frequently (*Figure 1C–D* and *Figure 1—figure supplement 1C–D*) during the reward-related cue (DS or CS+)

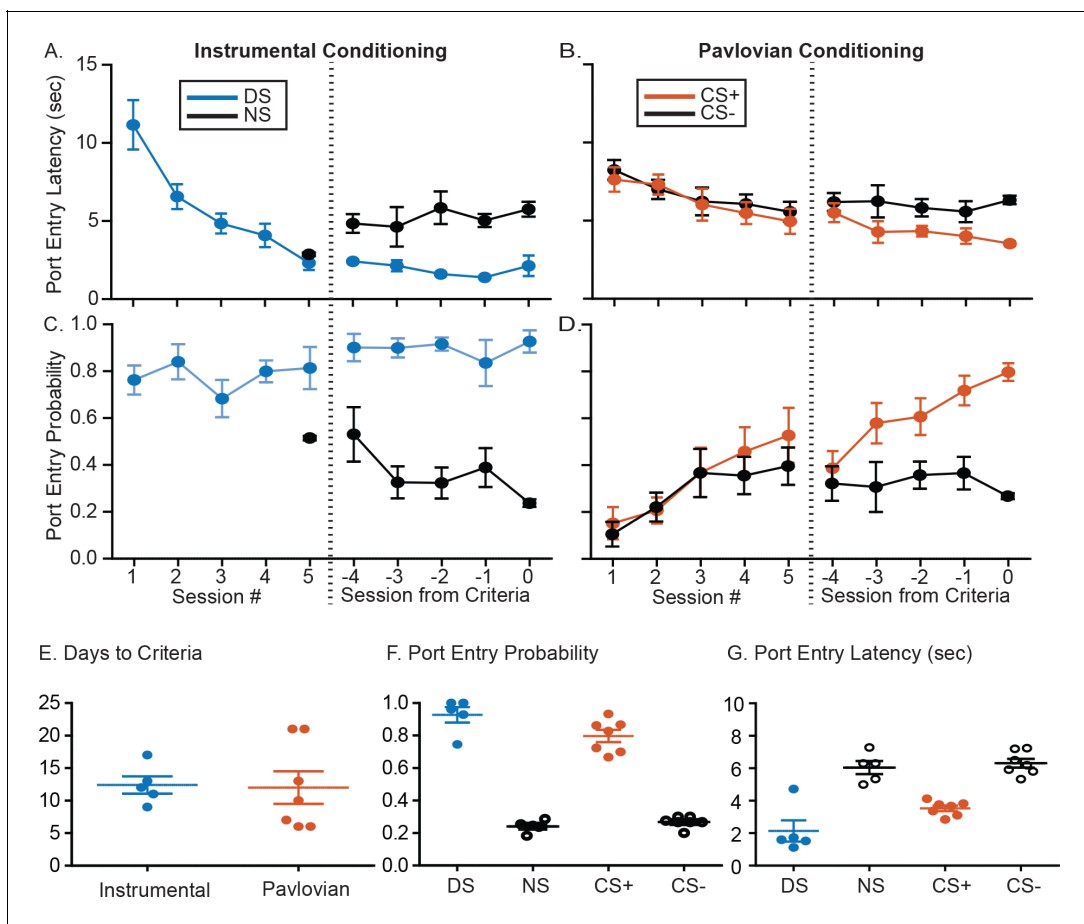

**Figure 1.** Summary of training behavior in the instrumental and Pavlovian tasks. Mean (± SEM) latency to enter the port during the instrumental DS (A, blue) or the Pavlovian CS+ (B, orange), and during the control cues (NS or CS-, black) during the initial 5 days of training and the last 5 days of training prior to meeting response criteria for electrode implantation. Mean (± SEM) probability of port entry during the instrumental DS (A, blue) or the Pavlovian CS+ (B, orange), and during the control cues (NS or CS-, black) during the initial 5 days of training and the last 5 days of training prior to meeting response criteria. Individual data points, and mean ± SEM for (E) days to reach response criteria, (F) final port entry probability, and (G) final port entry latency (sec) prior to electrode array implantation.

DOI: https://doi.org/10.7554/eLife.33107.003

The following figure supplement is available for figure 1:

**Figure supplement 1.** Training performance from individual rats prior to electrode implantation.

DOI: https://doi.org/10.7554/eLife.33107.004

than the control cue (NS or CS–). They were trained until they made port entries during at least 70% of reward cue presentations (DS or CS+) and less than 30% of control cue presentations (NS or CS–). Rats trained in the two tasks did not differ in the number of training days required to meet training criteria (*Figure 1E*; t(10)=0.1249, p=0.9031, Bayes factor of 2.112 in favor of the null). Once the rats met the training criteria, they were implanted with drivable electrode arrays aimed at the VP. Most recording sites were centered in middle to slightly caudal locations in VP (0 to 0.24 mm behind bregma), though one rat trained in the Pavlovian task had electrodes centered more caudally (~0.5 mm caudal to bregma). All comparisons between encoding in the instrumental and Pavlovian tasks were conducted with and without this subject, to ensure that our conclusions were not biased on the basis of recording location. The results reported here include this subject.

Overall, many VP neurons were responsive to cues, and at the time of port entry and/or reward delivery in both tasks (*Figure 2C*, *Figure 2—figure supplement 1A–D*, *Figure 3A and C*). Because we were primarily interested in determining the degree to which VP neurons encoded the learned or incentive value of cues in the two tasks, we focused the bulk of our analysis on explicit cue responses. The DS elicited increases in activity in 60% of VP units (189/314) and decreases in activity in 27% of units (85/314). A smaller, but still robust, population of VP neurons responded to the CS+, including 48.7% of VP neurons that were excited by the CS+ (191/392) and 20.9% of VP neurons that were inhibited (82/392). While the proportion of cue-responsive neurons differed across the two tasks (x2 = 30.34, p<0.001), we next assessed whether these cue responses encoded learned value or behavioral responses similarly.

## Ventral pallidal encoding of learned cue value is similar in the instrumental and Pavlovian tasks

To determine what information about the cues is encoded in the two tasks, we first assessed whether changes in VP neuron firing distinguished between the reward-related cue (DS or CS+) and the control cue (NS or CS–) in each task (*Figure 3A–D*). Most DS excited neurons were significantly more excited by the DS than the NS (*Figure 2—figure supplement 1A*; 132/189, 69.8%), whereas only 23.5% of DS-inhibited neurons were more inhibited by the DS than the NS (*Figure 2—figure supplement 1B*; 20/85, 6% of the whole population; *Figure 3B*). As a population, DS-excited neurons responded significantly more to the DS and the NS (*Figure 3E*; t[197]=15.43, p<0.001) and DS-inhibited neurons were more inhibited by the DS than by the NS (*Figure 3G*; t[84]=–3.10, p=0.0027), suggesting that both response types encode cue value. VP neurons also encoded learned cue value in the Pavlovian task: the majority of CS+ excited neurons were more excited by the CS + than by the CS– (*Figure 2—figure supplement 1C*; 64%; 123/191; 31.4% of the population), though only a small fraction of CS+ inhibited neurons were more inhibited by the CS+ than by the CS– (8.3%; 7/84;~2% of the whole population; *Figure 3D* and *Figure 2—figure supplement 1D*). As a population, CS+ excited neurons were significantly more excited by the CS+ than by the CS– (*Figure 3G*; t[190]=18.6890, p<0.001) and CS+ inhibited neurons were also more inhibited by the CS+ than by the CS– (*Figure 2H*; t[83]=-–2.72, p=0.01), indicating that, like VP responses in the instrumental task, both excitations and inhibitions predict cue value after Pavlovian conditioning. To further assess the ability of post-cue VP neuron firing to predict cue identity, we conducted receiver operating characteristic (ROC) analyses to assess the detection of either the DS or the CS+ versus their respective control cues. For each unit, we calculated the area under the ROC curve (auROC); auROCs greater than five indicate units that had greater firing rates on reward cue than control cue trials. We found no difference in the population auROC distributions for the DS versus the CS+ for predicting cue identity (*Figure 3I*; t[704]=1.22, p=0.22; Bayes factor of 5.706 favoring the null), suggesting that VP firing encodes cue value equally in both tasks. Post-cue firing was significantly more predictive of cue identity than pre-cue firing for both the instrumental (t[313]=11.8213, p<0.001) and the Pavlovian task (t[391]=11.0872, p<0.001). While cue responses were equally predictive of cue identity for the instrumental and Pavlovian tasks, DS excited neurons were more likely to be inhibited by the control cue (the NS) than CS+ excited neurons were to be inhibited by the CS– (*Figure 3J*; x2 = 63.9923, p<0.001), perhaps suggesting a more active role for response inhibition by the neurons in the instrumental task.

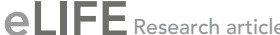

**Figure 2.** VP neurons respond to cues and reward seeking in both tasks. (A) Histological reconstruction of electrode array placements in the instrumental task, shown on coronal slices, marked relative to bregma (mm), with the boundaries of VP demarcated in red. Purple squares mark array locations contained within VP, and blue squares mark array locations that were not contained within VP. (B) Electrode array placements in the Pavlovian task. (C) Pie charts showing the proportion of neurons (labels indicated percentage of units) for which we detected

*Figure 2 continued on next page*

*Figure 2 continued*
an increase (yellow) or decrease (dark blue) in firing following the DS, CS+, post-DS port entry, post-CS+ port entry or post-CS+ reward delivery response windows.
DOI: https://doi.org/10.7554/eLife.33107.005
The following figure supplement is available for figure 2:

**Figure supplement 1.** Example VP cue responses after instrumental and Pavlovian training.
DOI: https://doi.org/10.7554/eLife.33107.006

### Ventral pallidum responses to cues more strongly predict the likelihood of instrumental versus Pavlovian behaviors, even when the behaviors are superficially similar

Beyond representing the predictive value of the DS, VP neurons have been previously shown to encode both the likelihood and latency of an instrumental reward-seeking response (*Richard et al., 2016*). Here we assessed, whether VP cue responses would do so similarly for superficially similar behavioral responses to the DS and the CS+. First, we assessed the magnitude of VP cue responses on trials with and without a behavioral response from those sessions with a sufficient number of trials without a response to allow reliable assessment of the difference (at least five trials). Because responding in these two tasks was more likely to approach 100% than in our previous experiment (*Richard et al., 2016*), probably due to the simpler, pre-potent nature of the port entry responses as opposed to a lever press, this trial criterion was lower than previously used, which may have reduced our ability to detect significant differences at an individual neuron level. When assessed individually (*Figure 4A*), about 20% of DS-excited neurons are significantly more responsive to the DS when it is followed by a response (24/130) and 16% of DS-inhibited neurons are more responsive on trials with a response (14/85). Only ~9% of CS+ excited neurons (*Figure 4B*) were more excited on trials with a response (14/160) and ~5% of CS+ inhibited neurons are more responsive (3/66). As a population, DS-excited neurons (130 units) were significantly more excited on trials when the DS was followed by a response (*Figures 4E* and *3I*; t[129]=8.60, p<0.001). DS-inhibited neurons (67 units) were also more inhibited on trials with a response (*Figure 3F and I*; t[66]=–4.66, p<0.001). Similarly, CS + excited neurons (174 units) were more excited on trials with a response (*Figure 4G and J*; t[174] =10.08, p<0.001), though CS+ inhibited neurons (82 units) did not distinguish between trials with and without a response (*Figure 4H and J*; t[81]=.479, p=0.633; Bayes factor: null is 6.46 times more likely than the alternative).

To assess the predictive ability of VP neuron firing rates in each task, we ran receiver operating characteristic (ROC) analysis to assess whether post-cue firing could predict the likelihood of a reward-seeking response, and then compared the distribution of auROCs for each task. Post-DS firing was significantly more predictive of port entry likelihood than post-CS+ firing (*Figure 4C and D*; t[704]=3.49, p<0.001), though post-cue firing was predictive of port entry likelihood for both the instrumental task (t[313]=3.708, p<0.001), and the Pavlovian task (t[391]=2.697, p=0.0073). Overall, post-cue firing in both tasks is predictive of reward seeking likelihood, but at least to some degree, DS responses are more robustly predictive of port entry likelihood than responses to the CS+.

### Ventral pallidum cue responses robustly predict the latency of instrumental but not Pavlovian reward-seeking behavior

Given the differences in the VP encoding of response likelihood in the two tasks, we next wanted to examine whether post-cue response differentially predicted response vigor in the two tasks. To do so, we examined whether cue responses on a given trial were predictive of the latency of the animal to respond on that trial, and whether this differed for instrumental versus Pavlovian port entries, by running Spearman rank correlations on individual neurons. The post-DS firing rate of 62 units (19.75%) significantly predicted the animal's latency to make a port entry (*Figure 5A*). By contrast, only 14 units (3.57%) had post-CS+ firing rates that significantly predicted port entry latency (*Figure 5C*), a much lower proportion of the population (DS versus CS+ % correlated: x2 = 47.477, p<0.001). To determine the degree to which the proportion of latency-predicting neurons in the Pavlovian task was meaningful, we ran the same analysis on 1000 shuffled iterations of the data and determined the number of units that significantly predicted response latency in these artificial



**Figure 3.** VP neural response are similarly predictive of the learned value of reward-related cues after instrumental or Pavlovian training. (**A**) Heatmaps of responses to the DS and NS from all neurons, sorted by the magnitude and direction of their response to the DS. Each line represents the peristimulus time histogram (PSTH) of an individual neuron, normalized (z-score) and color coded. (**B**) Scatterplot of normalized (mean z-score) responses to the DS versus the NS, showing neurons that fire significantly more following the DS (pink), the NS (green) or neither cue (black). Inset

*Figure 3 continued on next page*

*Figure 3 continued*

histogram shows the distribution of units based on the difference in their normalized response to the DS versus the NS. (C) Heatmaps of responses to the CS+ and CS– from all neurons, sorted by the magnitude and direction of their response to the CS+. Each line represents the PSTH of an individual neuron, normalized (z-score) and color coded. (B) Scatterplot of normalized (mean z-score) responses to the CS+ versus the CS–, showing neurons that fire significantly more following the CS+ (pink), the CS– (green) or neither cue (black). Inset histogram shows the distribution of units based on the difference in their normalized response to the CS+ versus the CS–. Average (mean ± SEM) normalized response to the DS (blue) and NS (black) in DS excited (E) or DS inhibited (F) neurons. Average (mean ± SEM) normalized response to the CS+ (orange) and CS– (black) in CS+ excited (G) or CS + inhibited (CS–) neurons. (I) Distribution of auROCs for the assessment of encoding of cue identity for the DS versus the NS (blue) and the CS+ versus the CS (orange). (J) Venn diagrams showing the degree of overlap between units with excitations or inhibitions to the DS and NS (left) or the CS+ and CS– (right). Numbers within each circle indicate the number of units that had a particular response pattern.
DOI: https://doi.org/10.7554/eLife.33107.007

datasets. The proportion of units with true post-DS firing that predicted the real latency was well outside the distribution of significantly correlated units from the shuffled datasets (*Figure 5—figure supplement 1E*), whereas the proportion of units in the Pavlovian task was not (*Figure 5—figure supplement 1F*). In addition, when we assessed the number of units with post-cue firing rates that were more predictive of the real latency than of the shuffled latencies (*Figure 5—figure supplement 1C and D*), we found only 9 units (2.8%) that predicted latency in the Pavlovian task, versus 61 in the instrumental task (19.42%). Furthermore, while latency-predicting neurons are, on average, significantly excited by presentations of the DS (*Figure 5D and G*; t[61]=7.08, p<0.001; 48/62 units excited, 77.4%), latency-predicting neurons in the CS+ task are not (*Figure 5E and H*; t[13]=2.00, p=0.06650; 7/14 units excited; Bayes factor of 1.27 in favor of the alternative). Latency-predicting neurons are more likely to be excited by the cue in the instrumental task than in the Pavlovian task (x2 = 4.294, p=0.038246).

For the population, the distribution of correlation coefficients was significantly more negative in the instrumental task than in the Pavlovian (*Figure 5A and C*; t[703]=2.0493, p=0.04), though the distribution is more negative in the 300 msec post-cue than in the 300 msec pre-cue in both the instrumental t[313]=–10.81, p<0.001) and Pavlovian tasks (t[390]=– 9.52, p<0.001). This suggests that while latency encoding in the Pavlovian task is much weaker, VP firing rates do encode some information relating to the subsequent port entry latency. To determine when this information is first encoded by VP neurons, we assessed correlation coefficients (*Figure 5F*) and the proportion of units with significant correlations (*Figure 5I and J*) in 50 msec windows starting from 0.5 s before to 1 s after cue onset. Surprisingly, we found that the distribution of correlation coefficients in the Pavlovian task shifted negatively in the 50 msec window prior to cue onset (*Figure 5F*; q=3.532, p<0.05), suggesting that the weak relationship between post-CS+ firing and latency may be accounted for by trial-by-trial variation in firing rates, rather than in phasic cue responses. By contrast, the distribution of correlation coefficients in the DS task did not significantly differ from 0 until the window 50–100 msec post-DS (*Figure 5F*; q=4.171, p<0.05), when phasic cue excitations are occurring in the bulk of correlated neurons.

Overall, encoding of the latency of cue-elicited reward seeking by individual VP units and in the population is much stronger in the instrumental task than in the Pavlovian task. These differences are not explained by greater variability in post-cue firing or in latency in the DS task. Variability in post-cue firing did not differ when comparing the whole population (*Figure 5—figure supplement 2A*; t [704]=–1.16, p=0.24; Bayes factor of 6.10 favoring the null) and was greater in the Pavlovian task when only the cue excited neurons were considered (*Figure 5—figure supplement 2B*; t[298]=2.75, p=0.0063). In addition, post-cue port entry latency is more variable in the Pavlovian task (*Figure 5—figure supplement 2C and D*; z(95354)=–5.82, p<0.001), indicating that weak correlations between post-CS+ firing and latency are not due to low variability in post CS+ port entry latency.

## Ventral pallidum encoding of reward-seeking latency is related to task engagement and not locomotor velocity or movement-onset latency

To further assess whether neural encoding differences could be accounted for by subtle locomotor differences between the two tasks, we conducted frame-by-frame video tracking analysis to assess velocity and distance from the reward port during the pre- and post-cue periods (*Figure 5—figure supplement 3A–H*), as well as the timing of post-cue movement onsets (*Figure 5—figure*

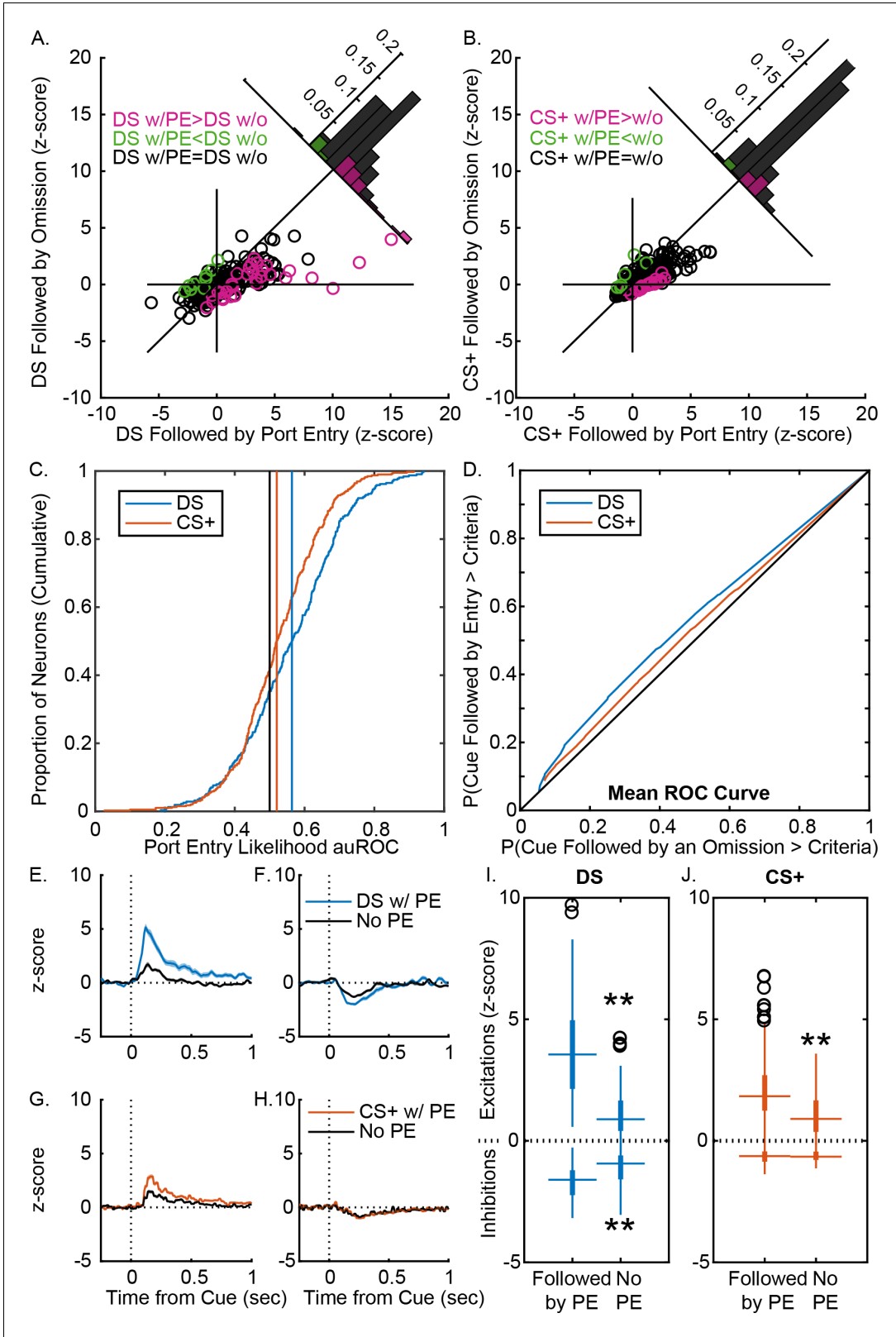

**Figure 4.** VP responses to cues more robustly predict response likelihood after instrumental training. (A) Scatterplot of normalized (mean z-score) responses to the DS on trials when it was followed by a port entry versus responses to the DS on trials when it was not followed by a port entry, showing neurons that fire significantly more on trials with a response (pink), on trials with no response (green) or on neither trial type (black). Inset histogram shows the distribution of units based on the difference in their normalized response on the two trial types. (B) Scatterplot of normalized

*Figure 4 continued on next page*

Figure 4 continued

(mean z-score) responses to the CS+ on trials when it was followed by a port entry versus responses to the CS+ on trials when it was not followed by a port entry, showing neurons that fire significantly more on trials with a response (pink), on trials with no response (green) or on neither trial type (black). Inset histogram shows the distribution of units based on the difference in their normalized response on the two trial types. (C) Distribution of auROCs for the assessment of response likelihood prediction following the DS (blue) or the CS+ (orange), with vertical lines showing the mean auROC for each distribution, and the mean from the control distribution (black). (D) Mean ROC curve for the DS (blue) and the CS+ (orange). Average (mean ± SEM) normalized response to the DS on trials with a port entry (blue) and without a port entry (black) in DS excited neurons (E) and DS inhibited neurons (F), shown from 0.5 s prior to 1 s after cue onset. Average (mean ± SEM) normalized response to the CS+ on trials with a port entry (orange) and without a port entry (black) in CS+ excited neurons (G) and CS+ inhibited neurons (H), shown from 0.5 s prior to 1 s after cue onset. (I) Box plots of normalized responses to the DS (blue) on trials with (left) and without (right) a port entry in DS excited (top) and DS inhibited (bottom) neurons. (J) Box plots of normalized responses to the CS+ (orange) on trials with (left) and without (right) a port entry in CS+ excited (top) and CS+ inhibited (bottom) neurons. **=p < 0.01.

DOI: https://doi.org/10.7554/eLife.33107.008

supplement 4) from a subset of sessions. Overall, although we found that velocity increased and distance from the port decreased shortly after cue onset for both tasks, instrumentally trained rats moved at greater velocities post-cue (F[1,118]=23.166, p<0.001) and were positioned at further distances from the port at cue onset (F[1,118]=6.99, p=0.009). We also assessed movement onsets to determine whether greater encoding of latency in instrumentally conditioned rats might be due to a greater incidence of movement onsets during the immediate post-cue period (*Figure 5—figure supplement 4*). Rats trained under Pavlovian versus instrumental contingencies did not differ in the likelihood of movement onset during the first 300 ms post cue in which we assessed neural firing (13.3% of post-DS movement onsets and 8.3% of post-CS+ movement onsets; $X_2$=0.345, p=0.56), and we did not observe a significant difference in movement-onset times (F[1,74]=1.365, p=0.24). To determine whether subtle differences in movement (velocity and movement onset) and/or non-movement (distance) variables drive differences in neural encoding between the two tasks, we assessed whether the relationships between post-cue firing and latency in individual neurons were predicted by the relationship between post-cue firing in these same units and trial-by-trial a) distance from port, b) post-cue velocity, or c) movement onset latency. We found that the only significant single predictor of latency encoding was distance encoding, in that neurons that had greater post-cue firing rates on trials with shorter port entry latencies, also had greater post-cue firing on trials where the rat was closer to the port at cue onset (*Figure 5—figure supplement 5A*; F[1,140]=16.295, p<0.001), similar to results previously reported for the nucleus accumbens (*McGinty et al., 2013*). This relationship between encoding of proximity and vigor depended on training history (F[1,140]=4.55, p=0.034), as distance encoding in the rats trained under a Pavlovian contingency was not significantly predictive of latency encoding (F[1,44]=2.09, p=0.15), whereas distance encoding robustly predicted latency encoding in instrumentally trained rats (F[1,96]=35.632, p<0.001). Notably, latency encoding was not significantly predicted by the degree to which neurons encoded post-cue velocity (*Figure 5—figure supplement 5B*; F[1,140]=–0.335, p=0.56) or movement-onset latencies (*Figure 5—figure supplement 5C*; F[1,140]=1.83, p=0.17). Proximity encoding has been suggested to reflect pre-cue variables such as attention or 'task engagement', but may also reflect encoding of other information derived from distance, such as expected effort (*Hamid et al., 2016*; *McGinty et al., 2013*; *Nicola, 2010*). That the relationship between distance and latency encoding was modulated by training history suggests a greater coupling between these variables and action in the instrumental task.

## VP inactivation increases the latency of cue-elicited reward seeking selectively in the instrumental task

Given that VP neurons differentially encode the likelihood and vigor of cue-elicited reward-seeking behavior, we next wanted to assess the functional contributions of VP activity to performance in these tasks. As in the electrophysiology experiments, we trained rats in either the DS task or in Pavlovian conditioning. By the end of training, rats made port entries at significantly shorter latencies during the reward cue than during the control cue (main effect of cue: F[1,28]=25.314, p<0.001), regardless of training group (interaction of cue and training: F[1,28]=0.144, p=0.707; *Figure 6—figure supplement 1A,B and F*). Rats in both training groups increased their probability of port entry preferentially during the reward-related cue versus the control cue (main effect of cue: F[1,28]

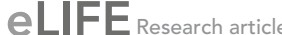

**Figure 5.** VP responses to cues predict response latency after instrumental, but not Pavlovian training. (**A**) Distribution of Spearman rank correlation coefficient relating firing (0–300 ms after DS onsets) to the rats' latency to enter the reward port on a trial-by-trial basis. Bars shaded in dark grey show neurons with significant correlations; bars shaded in light blue show neurons without significant correlations. (**B**). Example DS-excited neuron with a significant negative correlation between firing rate and latency. Individual trials are sorted by latency between the onset of the DS (blue) and the time

*Figure 5 continued on next page*

*Figure 5 continued*

over the port entry (green triangle). The red box indicates the time window during which firing rate was assessed. The rasters and corresponding histogram are aligned to DS onset (time 0). (C) Distribution of Spearman rank correlation coefficient relating firing (0–300 ms after CS+ onsets) to the rats' latency to enter the reward port on a trial-by-trial basis. Bars shaded in dark grey show neurons with significant correlations; bars shaded in orange show neurons without significant correlations. Heatmaps of responses to the DS (D) and CS+ (E) from all neurons with post-cue firing rates that significantly predict response latency, sorted by correlation coefficient. Each line represents the PSTH of an individual neuron, normalized (z-score) and color coded. Average (mean ± SEM) normalized responses are shown below the heatmaps for the DS (G) and CS+ (H). Mean (± SEM) correlation coefficients (F) during 50 ms windows, from 0.5 s to 1 s after the DS (blue) or the CS+ (orange). Proportion of neurons with firing rates that are significantly negatively (green) or positively (pink) correlated with latency to enter the port during 50 ms windows starting 0.5 s before 1 s after the DS (I) or the CS+ (J).

DOI: https://doi.org/10.7554/eLife.33107.009

The following figure supplements are available for figure 5:

**Figure supplement 1.** Characterizing correlations between DS and CS+ evoked firing and responses latency in comparison to shuffled data controls.
DOI: https://doi.org/10.7554/eLife.33107.010
**Figure supplement 2.** The absence of robust latency encoding in the Pavlovian task is not due to decreased variability in either post-cue firing or behavioral latency.
DOI: https://doi.org/10.7554/eLife.33107.011
**Figure supplement 3.** Analysis of velocity and distance from reward port.
DOI: https://doi.org/10.7554/eLife.33107.012
**Figure supplement 4.** Movement onsetdistributions.
DOI: https://doi.org/10.7554/eLife.33107.013
**Figure supplement 5.** Relationship between latency encoding and encoding of distance, velocity and movement onsets.
DOI: https://doi.org/10.7554/eLife.33107.014

=951.681, p<0.001), with rats trained under the instrumental contingency reaching a slightly higher response probability during the reward cue once they met training criteria (interaction of cue and training: F[1,28]=5.962, p=0.021; *Figure 6—figure supplement 1C,D and G*). On average, rats trained under the Pavlovian conditioning required more days to reach training criteria (10.35 ± 0.23) than those trained under an instrumental contingency (9.07 ± 0.24; t[29]=3.839, p<0.001; *Figure 6— figure supplement 1E*).

In the instrumental task, inactivation of VP with low doses of the GABA agonists baclofen and muscimol reduced the probability of port entry during the cue period (*Figure 6B* and *Figure 6—figure supplement 2B*; main effect of treatment: F[4,130]=28.666, p<0.001; Sidak, p<0.001), and did so more strongly during the DS period than during the NS (interaction between treatment and cue identity: F[4,130]=7.205, p<0.001; DS Sidak, p<0.001; NS Sidak, n.s.). VP inactivation also increased port entry latency during the cue (*Figure 6D* and *Figure 6—figure supplement 2A*; main effect of treatment: F[4,18.254]=4.574, p=0.01) also selectively during the DS (interaction between drug and cue: F[4,23.879]=4.354, p=0.009; DS Sidak, p=0.022; NS Sidak, n.s.). Inactivation of VP during the Pavlovian task similarly reduced the probability of port entry during the cue (*Figure 6C* and *Figure 6—figure supplement 3B*; main effect of treatment: F[4,112]=9.904, p<0.001; Sidak p<0.001), and did so selectively during the CS+ period (interaction between treatment and cue identity: F[4,112]=2.622, p=0.039; CS+ Sidak, p<0.001; CS- Sidak, n.s.). In contrast to the instrumental task, VP inactivation during the Pavlovian task had no significant effect on port entry latency (*Figure 6E* and *Figure 6—figure supplement 3A*; main effect of treatment: F[4,21.406]=1.059, p=0.401), regardless of cue identity (interaction between treatment and cue identity: F[4,18.061]=1.742, p=0.185). In both tasks, no drug treatment had any significant effect on port entry behavior during the inter-trial interval (main effect of treatment, instrumental task: F[4,65]=2.295, p=0.07; Pavlovian task: F[4,63]=1.526, p=0.205).

## Disrupting glutamate and dopamine, but not substance P, signaling in VP slows instrumental but not Pavlovian reward-seeking behaviors

VP receives a number of inputs that may contribute to reward-cue excitations that have not been previously studied, including glutamatergic inputs from a variety of cortical and cortical-like structures (*Fuller et al., 1987*; *Kelley et al., 1982*; *Maslowski-Cobuzzi and Napier, 1994*; *Reep and Winans, 1982*; *Záborszky et al., 1984*), as well as from the lateral hypothalamus (*Grove, 1988*),



**Figure 6.** VP inactivation or disruption of local glutamate or dopamine signaling impacts reward-seeking latency after instrumental, but not Pavlovian training. (A) Histological reconstruction of microinjection sites in the instrumental task (diamonds) and in Pavlovian (circles) experiments shown on coronal slices, marked relative to bregma (mm), with the boundaries of VP demarcated in red. Purple markers indicate microinjection locations contained within VP, and blue markers indicate microinjection locations that were not contained within VP. (B) Change from vehicle in the probability of

*Figure 6 continued on next page*

*Figure 6 continued*

a port entry during the CS+ (left) or CS– (right) in the instrumental task, following each drug treatment. (**C**) Change from vehicle in the probability of a port entry during the DS (left) or NS (right) in the Pavlovian task, following each drug treatment. (**D**) Change from vehicle in port entry latency during the CS+ (left) or CS– (right) in the instrumental task, following each drug treatment. (**E**) Change from vehicle in port entry latency during the DS (left) or NS (right) in the Pavlovian task, following each drug treatment. *, p<0.05, *p<0.01 pairwise comparison with Sidak corrections.

DOI: https://doi.org/10.7554/eLife.33107.015

The following figure supplements are available for figure 6:

**Figure supplement 1.** Summary of training behavior in the instrumental and Pavlovian tasks in the microinjection groups.

DOI: https://doi.org/10.7554/eLife.33107.016

**Figure supplement 2.** Maps of microinjection effects after instrumental training.

DOI: https://doi.org/10.7554/eLife.33107.017

**Figure supplement 3.** Maps of microinjection effects after Pavlovian training.

DOI: https://doi.org/10.7554/eLife.33107.018

midline thalamic nuclei, and subthalamic nucleus (*Fuller et al., 1987*), dopaminergic inputs from the midbrain (*Maslowski-Cobuzzi and Napier, 1994*; *Napier and Potter, 1989*) and substance P inputs from the nucleus accumbens (*Napier et al., 1995*). Therefore, we next sought to assess the contributions of glutamatergic, dopaminergic and substance P signaling in the VP to the likelihood and vigor of cue-elicited reward seeking after instrumental or Pavlovian training. Infusions of either a mixture of the glutamate receptor antagonists CNQX and MK801 or the relatively non-selective dopamine receptor antagonist flupenthixol had similar effects to VP inactivation with GABA agonists in both tasks. Glutamate or dopamine blockade reduced the probability of port entry during the cue (Sidak p<0.001 for both), though only the effect of glutamate blockade was selective to the DS (*Figure 6B* and *Figure 6—figure supplement 1B*; DS Sidak, p<0.001; NS Sidak, p=n .s.) whereas flupenthixol reduced the probability of port entry during both cue types (DS Sidak, p<0.001; NS Sidak, p<0.003). Glutamate or dopamine antagonism selectively increased port entry latency during the DS (CNQX-MK801 Sidak, p=0.005; flupenthixol Sidak, p=0.022), but not during the NS. In the Pavlovian task, glutamate or dopamine antagonism, like inactivation, significantly reduced port entry probability during the CS+ (*Figure 6C* and *Figure 6—figure supplement 2B*; CNQX-MK801 Sidak, p<0.001; flupenthixol Sidak, p=0.043), but not during the CS–, and had no effect on port entry latency (*Figure 6E* and *Figure 6—figure supplement 2A*). Infusions of the NK-1 antagonist WIN51708, used to block the effects of substance P, had no effect on port entry probability or latency in either task. Because, for many behaviors, VP functionality appears to be organized topographically (*Root et al., 2015*), we mapped our behavioral effects for each subject at each microinfusion site in the horizontal plane (*Figure 6—figure supplements 2* and *3*) to determine whether the null effects on latency in the Pavlovian task were due to mixed effects at distinct neuroanatomical locations. This does not appear to be the case because the vast majority of our infusion sites were located in the caudal half of VP, in and around the hedonic 'hot spot' (*Ho and Berridge, 2013*; *Smith and Berridge, 2005*; *2007*), and we found inconsistent changes in post-CS+ port entry latency even at this particular anatomical location (*Figure 6—figure supplement 3A*).

## Discussion

Here we demonstrate that while most VP neurons are responsive to reward-related cues in either a Pavlovian or instrumental task, the information encoded by these responses differs depending on the associative structure of the task. VP neuron responses to either a Pavlovian cue signaling reward delivery or a discriminative stimulus signaling reward availability are equally predictive of a cue's learned value, in that they selectively respond to the reward-related cue in comparison to its control cue. But, the degree to which VP cue responses predict the likelihood of a behavioral response are comparatively weak in the Pavlovian task. In addition, while the firing of many cue-excited VP neurons is predictive of response latency on a trial-by-trial basis, in the Pavlovian task, the proportion of VP neurons with firing rates at cue onset that predict response latency does not exceed chance levels. Further, we show that while activity in VP functionally contributes to normal response latency in the instrumental task, it does not in the Pavlovian task. Finally, we demonstrate that normal response latency in the instrumental task depends on local dopamine and glutamate signaling, but not on

substance P signaling, the main source of excitatory drive from the nucleus accumbens. Together, these results suggest that VP neuron firing that predicts and contributes to response latency in the instrumental task is not merely a motor invigoration signal, but more likely a neural instantiation of an underlying decision-making process that is reflected behaviorally by response latency in the instrumental task, but not the Pavlovian task.

## Do VP neurons drive a specific subset of motor behaviors?

Our results demonstrate that VP neurons do not drive movement invigoration generically, but encode a variable that is manifested by port entry latency after instrumental, but not after Pavlovian training. Yet, VP has not been classically construed as a motor output region (*Heimer et al., 1982*; *Mogenson et al., 1980*) without cause. Manipulations of VP alter a wide range of motor behaviors including general locomotion (*Churchill and Kalivas, 1999*; *Kitamura et al., 2001*; *Napier and Chrobak, 1992*; *Root et al., 2015*), though reports of locomotor activation linked to VP may be more related to activity in nearby structures such as the rostral preoptic area (*Zahm et al., 2014*). The differences in VP encoding of the latency of instrumental and Pavlovian approach behaviors that we report here are inconsistent with a general role in motor invigoration, but perhaps VP signaling serves to invigorate a specific subset of behavioral responses. For instance, similar cue-related signals in the nucleus accumbens and in midbrain dopamine neurons have been proposed to promote 'flexible approach', in which animals must navigate toward their goal location from a flexible starting location (*McGinty et al., 2013*; *Nicola, 2010*). It is unlikely, however, that promotion of flexible approach accounts for the differences in encoding that we report here for superficially similar approach behaviors. VP neuron activity may contribute to at least some forms of flexible approach, but only when that approach reflects a specific underlying decision variable.

## What decision variables are encoded by latency-predicting VP neurons?

VP activity has been suggested to signal many different decision-making variables including action values or state values (*Ito and Doya, 2009*). One critical difference between cue responses in the instrumental versus Pavlovian tasks may be the degree to which state values are linked to action values (*Averbeck and Costa, 2017*). While a stronger link between state and action values in the instrumental task may explain a more robust relationship between VP cue responses and reward-seeking behaviors, the existing literature does not make clear predictions about how changes in state value should alter response latency in these two conditioning paradigms, or about how response latency might be altered separately from response probability after Pavlovian conditioning. An alternative variable that predicts a more specific relationship between VP activity and the animals' reward-seeking vigor or motivation is incentive value, or the degree to which cues have the ability to activate motivational states (*Bindra, 1978*; *Robinson et al., 2014*), including those that invigorate ongoing reward-seeking actions. We and others have shown that this property of incentive cues requires normal activity in VP neurons (*Leung and Balleine, 2015*; *Prasad and McNally, 2016*; *Richard et al., 2016*).

Our finding that latency-encoding neurons are more likely to also encode proximity in instrumentally trained rats supports the hypothesis that these firing patterns reflect an underlying variable that is related to both proximity and response vigor. Proximity-encoding signals have been hypothesized to reflect encoding of information derived from distance, such as reward expectancy or expected effort, but may also be influenced by pre-cue factors such as 'task engagement' or motivational state at cue onset (*Howe et al., 2013*; *McGinty et al., 2013*; *Nicola, 2010*), consistent with an incentive motivational role for this firing (*Ahrens et al., 2016*; *Richard et al., 2016*; *Smith et al., 2011*). Within this framework, VP neuron activity may represent the incentive value of both the instrumental DS and the Pavlovian CS+, but the latency to approach the reward location following Pavlovian cues does not reflect this incentive value (*Ahrens et al., 2016*; *Chang et al., 2015*; *Robinson and Flagel, 2009*). This is supported by our finding that proximity encoding is similar in the two tasks, but is only correlated with latency encoding in the instrumental task. Alternatively, incentive value may be low in the Pavlovian task, and therefore the corresponding encoding less apparent, consistent with weaker VP population-level representation of reward cues in this task in general. We should note that VP activity both encodes and contributes to the likelihood of behavioral responses following the Pavlovian cue, though this encoding is weaker than in the instrumental task. VP activity may

contribute to response likelihood in the Pavlovian task via the same underlying circuits and processes as in the instrumental task, or through distinct neural and psychological mechanisms, such as then-coding of incentive value versus expected reward value (*Chan et al., 2016*; *Tindell et al., 2009*), as some evidence suggests that reward value and motivation rely on dissociable neural mechanisms even within VP (*Creed et al., 2016*).

## Functional heterogeneity along the rostrocaudal axis of VP: neural encoding of response vigor

Because rostral and caudal VP neurons have different morphologic and electrophysiologic properties (*Bengtson and Osborne, 2000*; *Kupchik and Kalivas, 2013*), and differentially modulate reward-related behaviors (*Chang et al., 2017*; *Ho and Berridge, 2014*; *Johnson et al., 1993*; *Mahler et al., 2014*; *McBride et al., 1999*; *Panagis et al., 1995*; *Smith and Berridge, 2005*), we previously assessed neural responses in the DS task in rostral, middle and caudal subregions of VP (*Richard et al., 2016*). We observed similar population encoding at rostral, middle and caudal recording sites in VP, in that most neurons in each subregion exhibited excitations following the DS, and post-cue firing in 18–25% of neurons predicted response latency. Although neurons throughout VP appear to encode cue value and the subsequent reward seeking similarly across the rostrocaudal axis, discrete populations may be differentially involved in dissociable behavioral responses to these reward cues (*Milton and Everitt, 2010*), including their ability to act as conditioned reinforcers (*Mahler et al., 2014*; *Torregrossa and Kalivas, 2008*) or to generate instrumental reward-seeking or other reward-related motivational states (*Leung and Balleine, 2013*). Here, we aimed to reduce the potential influence of subregional heterogeneity on differences in encoding or functional effects between the two tasks by focusing our recording and injections in the same middle to slightly caudal area of VP, in and around the VP 'hot spot'. Whether the differences reported here hold at more rostral regions of VP, remains an open question.

## Neural circuit mechanisms underlying the generation of VP incentive signals

VP has previously been conceptualized primarily as a major output structure of the nucleus accumbens, which has been demonstrated to encode the value of reward-predictive cues (*Ambroggi et al., 2008*, *2011*; *Day et al., 2006*) as well as both proximity to the response operandum and the vigor of subsequent reward-seeking actions (*McGinty et al., 2013*; *Nicola et al., 2004*). We previously showed that the timing of nucleus accumbens and VP cue responses is inconsistent with the idea that VP cue responses are a reflection of upstream activity changes in the accumbens (*Richard et al., 2016*). In addition, chemogenetic 'disconnection' of VP and accumbens during Pavlovian conditioning results in elevated sign-tracking behavior, suggesting a competitive interaction between these two sites (*Chang et al., 2018*). Since nucleus accumbens inputs to VP are primarily GABAergic, the likeliest mechanism by which accumbens activity changes could result in VP excitations is via disinhibition: yet we showed previously that post-cue excitations in VP neurons occur well before accumbens inhibitions. Further, we show here that blocking the actions of substance P, the main known source of excitatory drive from the accumbens to the VP (*Napier et al., 1995*), has no effect on the likelihood or latency of cue-elicited reward seeking.

By contrast, local disruption of either glutamate or dopamine signaling was effective in reducing the likelihood and speed of cue-elicited reward seeking, indicating a role for both neurotransmitter systems in VP contributions to the invigoration of reward seeking. VP neurons integrate glutamatergic inputs from a variety of brain areas implicated in reward learning and cue-elicited motivation (*Fuller et al., 1987*; *Grove, 1988*; *Kelley et al., 1982*; *Maslowski-Cobuzzi and Napier, 1994*; *Reep and Winans, 1982*; *Záborszky et al., 1984*), including the medial pre-frontal cortex (mPFC) (*Capriles et al., 2003*; *Ishikawa et al., 2008a*, *2008b*; *Moorman and Aston-Jones, 2015*; *Peters et al., 2009*; *Stefanik et al., 2013*) and the basolateral amygdala (BLA) (*Ambroggi et al., 2008*; *Ishikawa et al., 2008b*; *Jones et al., 2010a*, *2010b*; *McDonald, 1991*; *Perry and McNally, 2013*; *Záborszky et al., 1984*). Excitatory drive from the BLA onto VP neurons has been shown to be modulated by local dopamine following stimulation of the ventral tegmental area (VTA) (*Maslowski-Cobuzzi and Napier, 1994*; *Napier and Potter, 1989*). Importantly, inactivation of the VTA reduces the likelihood and increases the latency of instrumental responding (*Fischbach-Weiss et al.,*

2018), including DS-elicited reward seeking (*Yun et al., 2004*). The mechanisms by which dopamine and glutamate inputs might interact in VP to modulate response probability or vigor are an important area of future study.

## Conclusions

Here we demonstrate that VP neurons selectively encode and contribute to the vigor of cue-elicited reward-seeking actions when those actions are trained via an instrumental contingency, rather than via Pavlovian conditioning. These results indicate that VP encoding of vigor is not related to motor invigoration or reward expectancy per se, but to the ability of reward-related cues to invigorate actions upon which reward is contingent. Whether VP neurons signal the value of work or the incentive properties of cues more generally requires further investigation. Together, our results show that VP cue responses do not merely reflect motor invigoration, but encode a motivation signal that may be integrated in the VP.

# Materials and methods

## Subjects

Male and female Long Evans rats (n = 54; Envigo), weighing 250–275 grams at arrival, were individually housed in a temperature- and humidity-controlled colony room on a 12 hr light/dark cycle. Starting one day prior to the initiation of training in either task and until they met criteria for responding to the reward-related cue, rats were food restricted to ~18–20 g/rat/day, and the amount of food was adjusted daily to maintain rats at ~90% of their free-feeding body weight. All experimental procedures were approved by the Institutional Animal Care and Use Committee at Johns Hopkins University and were carried out in accordance with the guidelines on animal care and use of the National Institutes of Health of the United States.

## Pavlovian conditioning

Rats were randomly assigned one of the following two auditory cues as their conditioned stimulus (CS+) for training and testing: (1) white noise or (2) 2900 Hz tone. Rats received the alternate auditory cue as their CS–. During conditioning sessions, the CS+ and CS–, each lasting 10 s, were presented on a pseudorandom variable interval schedule with a mean inter-trial interval (ITI) of 50 s. At 8 s after the CS+ onset, 13 mL of 10% sucrose was delivered into the sucrose delivery port over a period of 2 s. Rats underwent daily conditioning until they met final response criteria (port entries on at least 70% of CS+ presentations and less than 30% of CS– presentations) prior to being implanted with electrode arrays or undergoing pharmacological manipulations of VP.

## DS task training

Rats were trained to perform a modified DS task, similar to those described previously (*Ghazizadeh et al., 2012*; *Richard et al., 2016*). Rats in the modified DS task group were randomly assigned one of the following two auditory cues as their DS for training and testing: (1) white noise or (2) a 2900 Hz tone. Rats received the alternate auditory cue as their NS (neutral stimulus). Entries into the sucrose delivery port during the DS presentation resulted in liquid sucrose (0.13 ml, 10%) delivery and termination of the DS cue. Port entries during the NS presentation or during the inter-trial interval (ITI) had no programmed consequences. Rats underwent the following sequential training stages (1) DS only, up to 60 s, (2) DS only, up to 30 s, (3) DS only, up to 20 s, (4) DS only up to 10 s, and (5) final stage, DS and NS (*Figure 1—figure supplement 1*). Once rats met preliminary DS response criteria at each stage (port entries on at least 60% of DS presentations), they were advanced to the next stage on the following day. During the final stage of the DS task, the DS and NS were presented on a pseudorandom variable interval schedule with a mean ITI of 50 s, and each trial lasted up to 10 s. Rats were trained to final criteria (port entries on at least 70% of DS presentations and less than 30% of NS presentations) prior to being implanted with electrode arrays or undergoing pharmacological manipulations of VP.

## Surgeries

During surgery, rats were anesthetized with isoflurane (5%) and placed in a stereotaxic apparatus, after which surgical anesthesia was maintained with isoflurane (0.5–2.0%). Rats received pre-operative injections of carprofen (5 mg/kg), topical lidocaine for analgesia and cefazolin (75 mg/kg) to prevent infection. Guide cannulae, electrodes and microdrives were secured to the skull with bone screws and dental acrylic. All rats were given at least 7 days to recover prior to any microinjections or tethering.

## Electrophysiological recordings

After they had reached response criteria in either task, rats in the electrophysiology experiments (n = 11; Pavlovian conditioning, n = 6; DS task, n = 5) received unilateral arrays of 16 electrodes each (0.004' tungsten wires arranged in a bundle) attached to microdrive devices that allowed the entire array to be lowered by 80 or 160 μm increments. Electrode arrays were targeted to VP at +0.0 mm AP, +2.4 mm ML, and −8.0 mm DV.

## Pharmacological manipulation of VP

Rats used in the pharmacology experiments received cannulae implants prior to training. Rats received bilateral 23-gauge guide cannulae (Plastics One, Roanoke, VA) aimed 2 mm above the VP (n = 43) using the following coordinates in comparison to bregma: 0.0 mm anteroposterior (AP), ±2.3 mm mediolateral (ML), –6.2 mm dorsoventral (DV). Wire obturators were inserted into guide cannulae and were flush with the ends of the guide cannulae to avoid occlusion.

## Electrophysiologal recordings

Electrophysiological recording was conducted as described previously (*Ambroggi et al., 2008*; *Ghazizadeh et al., 2010*; *2012*; *Nicola et al., 2004*; *Richard et al., 2016*). Rats were connected to the recording apparatus (Plexon Inc, TX), consisting of a head stage with operational amplifiers, cable and a commutator to allow free movement during recording. Rats were run for 1.5 hr daily sessions, and recording was started after rats regained performance criteria (port entries during at least 70% of CS+ or DS presentations and less than 30% of control cue presentations) following surgery. The microdrive carrying the electrode arrays was lowered by 80–160 μm at the end of each session with satisfactory behavior (>60% of DS or CS+ presentations with a response), in order to obtain a new set of neurons for each session that was included in the analysis.

## Analysis of electrophysiological recordings

### Spike sorting

Isolation of individual units was performed off-line with Offline Sorter (Plexon) using principal component analysis as described previously (*Ambroggi et al., 2008*; *Ghazizadeh et al., 2012*). Inter-spike interval distribution, cross-correlograms and autocorrelograms were used to ensure that single units were isolated. Only units with well-defined waveforms with characteristics that were constant over the entire recording session were included in the study. Sorted units were exported to Neuro-Explorer 3.0 and Matlab for further analysis.

### Determination of the optimal bin size

Optimal bin size was determined as described previously (*Ambroggi et al., 2011*; *Ghazizadeh et al., 2012*). Briefly, the optimal bin size for each neuron was found using Akaike Information Criteria (AIC). Because the optimal bin size is rather large for most neurons, and because the AIC shows a fast reduction over small bin sizes, followed by slow changes around the optimal bin size, we used the smallest possible bin size that showed less than a 10% change from the optimal AIC value. This bin size, referred to as the deflection point, was on average 50 ms across the population.

### Response detection

Peristimulus time histograms (PSTHs), constructed around the behavioral events using the optimal bin size, were used to detect the presence of excitations and inhibitions, as well as their onsets and

offsets, in comparison to a 10 s baseline period prior to cue onset. At least one bin outside of the 99% confidence interval of the baseline during the analysis window for each event was required to determine an excitation or inhibition for that event. Onset detection was performed using a telescopic approach: within the first bin outside of the 99% confidence interval, we searched for the first of three 10 ms bins in which firing was beyond the 99% confidence interval of the baseline based on a 10 ms bin size. This allowed us to determine onset times using the highest valid resolution for each PSTH. Response offset was identified by finding the start of the first bin after the detected onset in which firing fell within the 99% confidence interval for at least 300 ms. Because inhibitions are more difficult to detect using this method, we also assessed response direction by running Wilcoxon sign-rank tests on firing during the 300 ms post-cue versus firing during the 10 s baseline period; this enabled us to detect significant inhibitions in firing rate that could not be detected using the telescopic onset detection approach. To compare firing in response to different task events (e.g. DS versus NS), Wilcoxon rank-sum tests were run on event-related firing during these same windows. The firing rate of each neuron during each bin of the PSTH was transformed to a z-score as follows: $(F_i - F_{mean})/F_{sd}$, where $F_i$ is the firing rate of the $i$th bin of the PSTH, and $F_{mean}$ and $F_{sd}$ are the mean and the SD of the firing rate during the 10 s baseline period. Color-coded maps and average traces were constructed on the basis of these z-scores.

## Receiver operating characteristic (ROC) analysis

To assess the ability of VP neural firing to predict the presence or identity of cues and subsequent behavioral responses, we conducted receiver operating characteristic (ROC) analysis evaluating: (1) the DS or CS+ response window in comparison to baseline, (2) the DS window versus the NS window or (3) the CS+ window versus the CS– windows, and (4) cue presentations that are followed by a port entry versus cue presentations that are not followed by a port entry. For each analysis, we assessed the probability that firing during each window met criteria that ranged from zero to the maximum firing rate for that neuron, and plotted the true positive rate (the likelihood that the firing in the window of interest was above criteria) against the false positive rate (the likelihood that the firing in the control window was above criteria) to create a ROC curve for each neuron. We then assessed the area under the ROC curve (auROC) for all VP neurons as well as the average ROC curve for the whole population. We also conducted ROC analysis comparing two randomly selected baseline windows and then compared the test auROC distributions to this control auROC distribution.

## Correlation analysis and shuffled data controls

To assess the relationship between post-DS and post-CS+ firing rate and latency to enter the reward port in individual neurons, we ran Spearman's rank correlations. To assess the degree to which significant correlations occur spuriously in a small percentage of units, we ran Spearman correlations on 1,000 iterations of shuffled data. The trial-by-trial latency was randomly shuffled for each neuron for each iteration, and Spearman correlations that related latency and firing rate during the 90–300 ms post-cue window were assessed for each neuron. We then evaluated the distribution of both the number of units with significant correlations (at p<0.05) and the mean correlation coefficient, confirming that both the mean correlation coefficient and the number of significant units from the real data were well outside the shuffled data distribution. To assess the degree to which these correlations were cue locked, we assessed the relationship between firing rate and port entry latency during 50 msec windows beginning 0.5 s prior to the cue onset, and finishing 1 s after. We ran a one-way ANOVA on the correlation coefficients for these windows, using Dunnett's multiple comparisons to determine the bin during which the correlation coefficients first differed from baseline.

## Comparing population and rate encoding across the tasks

To assess differences in population encoding (i.e. the degree to which the proportions of neurons that responded to task events differed) across the two tasks, we used chi-squared ($X_2$) tests. To assess differences in the rate encoding of cue identity (auROCs), response likelihood (auROCs), response latency (correlation coefficients) or response variability (latency or firing rate standard deviations) between the two tasks, we used two sample t-tests or Wilcoxon rank sum tests, as appropriate. When the effect of task had a critical non-significant effect, we followed this analysis by calculating a Bayes factor to assess support for the null hypothesis (Scaled JZS Bayes Factor).

## Video data acquisition, processing and analysis

For a subset of recording sessions, overhead video was captured at 30 frames per s (*Figure 5—figure supplement 1A and B*). For those subjects for which video was available (n = 4, 2 instrumental and 2 Pavlovian), Noldus Ethovision software was used to track the position of a red LED mounted on the recording headstage located on top of the rats' heads during the session, with the maximum number of recorded units located in VP for each subject. The position of the LEDs was transformed into a position coordinate, with missing data points filled in using linear interpolation. An experimenter who was blind to conditioning group viewed all pre- and post-cue periods to adjust for errors in tracking or interpolation manually. These position coordinates were used to determine frame-by-frame velocity (cm/s) and distance from the reward port (cm) during the pre- and post-cue periods (*Figure 51—figure supplement 1C–H*). Velocity data were used to determine the latency of movement onsets following each cue presentation, by finding the first of at least five consecutive frames with velocity values greater than the 95% confidence interval of baseline velocity (5 s pre-cue; *Figure 5—figure supplement 2*). To assess differences in these variables across the two tasks, we fit linear mixed-effects models for each movement variable with a fixed effect for training history and a random effect for subject. To determine the degree to which neural encoding of port entry latency was related to encoding of these response characteristics, we first computed Spearman's rank correlation coefficients for each unit and trial-by-trial velocity (0 to 0.5 s and 0.5 to 1 s post-cue, cm/s), distance from the port (cm) and movement onset latency (s). Then, we fit a linear mixed-effects model of the correlation coefficients relating neural firing to port entry latency, with fixed effects for training history and the correlation coefficients for velocity, distance and movement onset latencies, as well as a random effect for subject.

## Intracranial microinjections

Drug microinjections were administered bilaterally in a 0.3 µl volume on test days spaced at least 48 hr apart, and counterbalanced for drug order across rats. On test days, solutions were brought to room temperature, and were infused at a rate of 0.3 µl per minute using a syringe pump attached via PE-20 tubing to stainless steel injectors (Plastics One, 29-gauge) that extended 2 mm beyond the end of the guide cannulae into VP. Injectors were left in place for 1 min to allow diffusion of the test solution, after which the experimenter replaced the obturators and immediately placed the rat in the testing chamber to begin the session. After reaching criteria in either task and prior to test sessions, rats received microinjections of vehicle immediately prior to a final training session, to habituate them to the testing procedure. Following each test day, rats received at least one drug-free retraining session to ensure that initial performance criteria were met.

## Drugs and solutions

In order to test the importance of VP activity and the role of specific neurotransmitter systems in cue-elicited behavior, rats (n = 43; Pavlovian conditioning, n = 23; DS task, n = 20) received bilateral VP infusions of the following drugs (amounts given are per hemisphere): a mixture of the GABA$_A$ agonist muscimol and the GABA$_B$ agonist baclofen (10 ng each in 0.3 µl 0.15 M saline) to inactivate VP; a mixture of the AMPA/kainate receptor antagonist CNQX (450 ng) and the NMDA receptor antagonist MK-801 (2 µg in 0.3 µl 90% saline/10% DMSO vehicle); the relatively non-selective dopamine receptor antagonist, flupenthixol (15 µg in 0.3 µl saline); and the NK-1 receptor antagonist WIN57108 (10 ng in 0.3 µl 90% saline/10% DMSO vehicle) to block the actions of substance P. Because WIN51708 and CNQX were initially dissolved in a small amount of DMSO, we also compared the effects of saline vehicle with the 90% saline/10% DMSO mixture to ensure that there were no differing effects of the vehicle solutions themselves, and found no significant effects (all ps > 0.05).

## Analysis of microinjection effects

On the basis of histological analysis, a total of 31 rats were included in the analysis of the microinjection effects, 17 from the Pavlovian task and 14 from the instrumental task. Effect sizes were not determined a priori, but we aimed to include at least 12 subjects with accurate bilateral cannulae placements in each behavioral group, based on established standards. To compare the effects of VP inactivation or receptor blockade on responses to Pavlovian versus instrumental cues, we primarily

focused our analysis on the likelihood and latency of port entry during each cue type. To account for some missing data (e.g. when port entry probability went to zero, no latency data were available), we analyzed the data using linear mixed models (drug X cue type [CS+ versus CS- *or* DS versus NS]), followed by pairwise comparisons with Sidak corrections. To achieve this, we used Akaike's information criteria to determine the best-fitting covariance model. Depending on the best-fitting covariance model (*Verbeke and Molenberghs, 2009*), the degrees of freedom may be a non-integer value. In both tasks, the best fitting model for the assessment of latency was a first-order antedependence model and the best fitting model for the assessment of probability was an identity model. We also analyzed the number of port entries during the inter-trial interval using linear mixed models with an identity covariance structure, followed by pairwise comparisons with Sidak corrections.

## Histology

Animals were deeply anesthetized with pentobarbital and electrode sites were labeled by passing a DC current through each electrode. All rats were perfused intracardially with 0.9% saline following by 4% paraformaldehyde. Brains were removed, post-fixed in 4% paraformaldehyde for 4–24 hr, cryoprotected in 25% sucrose for >48 hr, and sectioned at 50 um on a microtome. We verified the location of injection or recording sites using two methods. One set of tissue was stained with cresyl violet and analyzed using light microscopy. On the alternating set of sections, we performed immunohistochemistry for substance P (SP) to better demarcate the boundaries of VP. Sections were washed in PBS with bovine serum albumin and triton (PBST) for 20 min, and incubated in 10% normal donkey serum in PBST for 30 min, before incubating in primary antibody (rabbit anti-SP 1:6500 Immunostar #20064, RRID: AB_572266) in PBST overnight at room temperature. Sections were then washed with PBST three times, incubated in 2% normal donkey serum in PBS for 10 min, and incubated for 2 hr in secondary antibody in PBS (Alexa Fluor 594 donkey anti-rabbit 1:500 Thermo Fisher #A21207, RRID:AB_141637). Sections were then washed with PBS three times, mounted on coated glass slides in PBS, air-dried, coverslipped with Vectashield mounting medium with DAPI, and imaged on a fluorescent microscope. The dorsoventral location of recording sites was determined by subtracting the distance driven between recording sessions from the final recording location. Units that were recorded during sessions when the recording sites were determined to be localized outside of the VP were excluded from analysis.

## Acknowledgements

This work was supported by National Institutes of Health grants F32 AA022290 (JMR) and R01 AA014925 (PHJ), and by a NARSAD Young Investigator Award (JMR). The authors thank Alexandra Haimbaugh for technical assistance and Karen Wang for assistance with behavioral video tracking.

## Additional information

### Funding

| Funder | Grant reference number | Author |
| --- | --- | --- |
| National Institute on Alcohol Abuse and Alcoholism | AA022290 | Jocelyn M Richard |
| Brain and Behavior Research Foundation | | Jocelyn M Richard |
| National Institute on Alcohol Abuse and Alcoholism | AA014925 | Patricia Janak |

The funders had no role in study design, data collection and interpretation, or the decision to submit the work for publication.

### Author contributions

Jocelyn M Richard, Conceptualization, Formal analysis, Funding acquisition, Investigation, Visualization, Writing—original draft, Writing—review and editing; Nakura Stout, Deanna Acs, Formal

analysis, Investigation, Writing—review and editing; Patricia H Janak, Conceptualization, Resources, Funding acquisition, Writing—original draft, Writing—review and editing

### Author ORCIDs
Jocelyn M Richard  http://orcid.org/0000-0001-5750-0418
Patricia H Janak  http://orcid.org/0000-0002-3333-9049

### Ethics
Animal experimentation: All experimental procedures were approved by the Institutional Animal Care and Use Committee at Johns Hopkins University and were carried out in accordance with approved protocol RA14A320 and the guidelines on animal care and use of the National Institutes of Health of the United States.

### Decision letter and Author response
Decision letter https://doi.org/10.7554/eLife.33107.024
Author response https://doi.org/10.7554/eLife.33107.025

# Additional files

### Supplementary files
• Transparent reporting form
DOI: https://doi.org/10.7554/eLife.33107.019

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
