## [Decision Letter]

Thank you for submitting your article "Ventral pallidal encoding of reward seeking depends on the underlying associative structure" for consideration by *eLife*. Your article has been reviewed by three peer reviewers, one of whom is a member of our Board of Reviewing Editors and the evaluation has been overseen by Sabine Kastner as the Senior Editor. The reviewers have opted to remain anonymous.

The reviewers have discussed the reviews with one another and the Reviewing Editor has drafted this decision to help you prepare a revised submission.

Summary:

This study examined the role of the ventral pallidium (VP) in encoding reward-seeking behaviors. The authors used a carefully designed pair of tasks, one involving Pavlovian conditioning and the other involving instrumental learning, to compare VP's role in both situations, controlling for visuo-motor factors (e.g., the same sensory cues and the same superficial behaviors). Results from electrophysiology and inactivation studies show converging evidence that VP encodes motivational incentives in both situations, but dissociable relationships with behavior. Specifically, VP inactivation leads to lower likelihood and higher latencies of reward approach in instrumental conditioning, but only leads to lower likelihood of approach (but no higher latency) in Pavlovian conditioning.

The reviewers agreed that the study was well designed, conducted, and analyzed. There was also general enthusiasm for the idea of better understanding the role of the VP in learning and for distinguishing Pavlovian and instrumental learning.

However, the reviewers also agreed that a number of essential revisions are necessary, as detailed below.

Essential revisions:

1) The interpretation of the results rests strongly on the claim that behavior was essentially equivalent on the two tasks. However, this equivalence may be only superficial. It is important to show that the relatively subtle differences in neural activity and pharmacological effects on behavior for the two tasks types did not reflect the relatively subtle differences in behavior for the two tasks. This issue is important because it goes to the heart of how to interpret the rest of the data. Even if port entry probability and latency do not differ, these are more blunt tools than those employed for the analysis of neural data. For example, there could be differences in locomotor speed to approach the port that might be detectable with a set of beam breaks. There may be more subtle differences in consummatory behavior that could be detected by EMG analyses.

Consider, for example, the following two claims from the Abstract:

a) "We find that cue elicited activity in many VP neurons predicts the latency of instrumental reward seeking, but not Pavlovian response latency." The distributions of latencies were different for the two tasks. Could that difference have accounted for this result? For example, port-entry latencies tended to be shorter in the instrumental case. It therefore seems possible that this effect is because there are more movement-related responses in the time window tested in that case. Likewise, this relationship between VP activity and response latency on the instrumental task stays consistent 0-1 sec after cue onset (Figure 5), despite the fact that the neural response is only strong right at the beginning of that epoch. How statistically reliable are the correlations computed with low spike counts (including for those measured before cue onset)?

b) "Further, disruption of VP signaling increases the latency of instrumental but not Pavlovian reward seeking." Because the latency was tighter in instrumental conditions, is it possible that this condition was simply easier to disrupt (or at the very least easier to tell statistically if it was disrupted)?

Given that the reported behavioral similarities and differences between the two tasks was based on a relatively small number of animals, it seems like it would be useful to show the same analysis of behavioral data as shown in Figure 1 for the 43 additional animals tested in the microinfusion studies.

2) It appears as though the history of reward for the two tasks is different, which in principle could lead to different learned valuation of the cues or actions taken by the animals. Can the authors discuss, or control for this in their analyses?

3) The authors describe the results in terms of incentive and conclude as to what the VP does not do (general motor envigoration) but remain extremely vague as to what they propose the VP does do ("encode a motivational signal that may be uniquely integrated in the VP"). Even the description in terms of incentive/motivational signal is a bit problematic – how could this produce distinct effects on reaction times (effects absent) from effects on choice (port entry – effects present) in the Pavlovian. The authors should ground their results in a theoretical proposal and discuss how their results support this theory in the discussion.

---

## [Author Response]

Essential revisions:1) The interpretation of the results rests strongly on the claim that behavior was essentially equivalent on the two tasks. However, this equivalence may be only superficial. It is important to show that the relatively subtle differences in neural activity and pharmacological effects on behavior for the two tasks types did not reflect the relatively subtle differences in behavior for the two tasks. This issue is important because it goes to the heart of how to interpret the rest of the data. Even if port entry probability and latency do not differ, these are more blunt tools than those employed for the analysis of neural data. For example, there could be differences in locomotor speed to approach the port that might be detectable with a set of beam breaks. There may be more subtle differences in consummatory behavior that could be detected by EMG analyses.Consider, for example, the following two claims from the Abstract:a) "We find that cue elicited activity in many VP neurons predicts the latency of instrumental reward seeking, but not Pavlovian response latency." The distributions of latencies were different for the two tasks. Could that difference have accounted for this result? For example, port-entry latencies tended to be shorter in the instrumental case. It therefore seems possible that this effect is because there are more movement-related responses in the time window tested in that case.

We agree with the reviewers that it is critical to assess the impact of subtle differences in locomotor behavior on the differences in neural encoding reported here. Overall, we think it unlikely that encoding of response latency in the instrumental task is specific to the movement response characteristics of the instrumental task used here, as we find very similar latency encoding using a version of the DS task in which rats were trained to make an entirely different operant response (lever press) as reported previously (Richard et al., 2016). Further, though “port-entry latencies tended to be shorter in the instrumental case”, short latency port entries did occur after the Pavlovian CS+ and were not preceded by greater post-cue firing than port entries that occurred at longer latencies (post-cue VP firing, training x port entry latency interaction: F(17,6500)=3.9765, p < 0.001).

To further assess whether neural encoding differences “reflect the relatively subtle differences in behavior for the two tasks” such as “differences in locomotor speed to approach the port”, as well as to determine whether “there are more movement-related responses in the time window tested” in instrumentally trained rats, we conducted detailed behavioral tracking analysis on videos that were obtained from a subset of subjects from the electrophysiological experiments (new Figure 5—figure supplement 3). Based on this analysis we assessed velocity and distance from the reward port during the pre- and post-cue periods (new Figure 5—figure supplement 3), as well as the timing of post-cue movement onsets. The details of this analysis are now described in the Methods section of the revised manuscript. Overall, while we found that velocity and distance from the port changed on a similar timescale for both tasks, instrumentally-trained rats moved at greater velocities post-cue (starting 0-.5 sec, F(1,118)=23.166, p < 0.001) and on average were positioned at further distances from the port at cue onset (F(1,118)=6.99, p = 0.009). We also assessed movement onsets to determine whether the correlation of cue-evoked firing with latency in instrumentally conditioned rats might be due to a greater incidence of movement onsets during the immediate post-cue period (new Figure 5—figure supplement 4). Rats trained under Pavlovian versus instrumental contingencies did not differ in the likelihood of movement onset during the first 300 msec post cue in which we assessed neural firing (13.3% of post-DS movement onsets and 8.3% of post-CS+ movement onsets; Χ2=.345, p = 0.56), and we did not observe a significant difference in movement onset times (F(1,74)=1.365, p = 0.24).

Yet the critical question remains whether these subtle differences in movement (velocity and movement onset) and/or non-movement (distance) variables drive differences in neural encoding between the two tasks. To answer this question, we assessed whether the relationships between post-cue firing and latency in individual neurons were predicted by the relationship between post-cue firing in these same units and trial-by-trial (a) distance from port, (b) post-cue velocity, or (c) movement onset latency, by fitting a linear mixed effects model that included these factors, along with a fixed effect for training and a random effect for subject. We found that the only significant single predictor of latency encoding was distance encoding, in that neurons that had greater post-cue firing rates on trials with shorter port entry latencies, also had greater post-cue firing on trials where the rat was closer to the port at cue onset (new Figure 5—figure supplement 5(1,140)=16.295, p < 0.001). A relationship between neural firing and distance from the port suggests that these neurons fire more to the cue when the rat is more “engaged” in the task, consistent with an incentive motivational role for this firing. The relationship between distance and latency encoding was modulated by training history (F(1,140)=4.55, p = 0.034): distance encoding in the rats trained under a Pavlovian contingency was not significantly predictive of latency encoding (F(1,44)=2.09, p = 0.15), whereas distance encoding robustly predicted latency encoding in instrumentally trained rats (F(1,96)=35.632, p < 0.001). This suggests a greater coupling of task engagement and action in the instrumental task than the Pavlovian. Latency encoding was not significantly predicted by the degree to which a neuron’s firing predicted post-cue velocity (new Figure 5—figure supplement 5(1,140)=-.335, p = 0.56) or movement onsets (new Figure 5—figure supplement 5(1,140)=1.83, p = 0.17). The results of these analyses have been integrated into the Results and Discussion sections.

Likewise, this relationship between VP activity and response latency on the instrumental task stays consistent 0-1 sec after cue onset (Figure 5), despite the fact that the neural response is only strong right at the beginning of that epoch. How statistically reliable are the correlations computed with low spike counts (including for those measured before cue onset)?

To address this question, we computed the correlations (blue line) between correlations computed in the larger response window 300 ms post-DS (blue shaded area) that were used for our primary analysis of latency encoding, with correlations computed in the smaller 50 ms windows (x-axis), which we utilized to examine the time course of latency encoding. This allowed us to assess whether the correlations in each bin were occurring in the same neurons whose phasic DS responses predicted port entry latency. Correlations computed based on the 300 ms post-DS were robustly correlated with those computed in smaller 50 ms bins within the 300 ms period, suggesting that these correlations from smaller bins (and therefore lower spike counts) are reliable indicators of the overall relationship between firing and latency. The decrease in correlation strength, and loss of significance 600 ms post-cue suggests that neurons with correlated activity at these later timepoints (after movement onsets) are not necessarily the same population that have immediate post-cue firing that predicts response latency. Because these later cue correlations are occurring during the bulk of movement initiation and locomotor activity, they are more likely to be contaminated by movement artifacts and/or responses related to port entry itself, which is why we focused our analysis on firing during phasic cue response prior to the bulk of movement onsets.

**Author response image 2. respfig2:** 

b) "Further, disruption of VP signaling increases the latency of instrumental but not Pavlovian reward seeking." Because the latency was tighter in instrumental conditions, is it possible that this condition was simply easier to disrupt (or at the very least easier to tell statistically if it was disrupted)?Given that the reported behavioral similarities and differences between the two tasks was based on a relatively small number of animals, it seems like it would be useful to show the same analysis of behavioral data as shown in Figure 1 for the 43 additional animals tested in the microinfusion studies.

We have also plotted and analyzed the behavioral training data from the microinjection study, as suggested by the reviewer, and include these data in a new supplemental figure (Figure 6—figure supplement 1). Overall, the training trajectory and differences for the two tasks was similar that of the electrophysiology groups, and while there were differences in average latency across the two tasks, we argue that these differences are not responsible for the selective functional effects for the following reasons. First, the lack of effect on latency after Pavlovian conditioning is not merely a problem of statistical detection, as more than half of the rats tested decreased their latency following VP inactivation (the opposite of the expected direction) and Bayes analysis suggests that the null hypothesis was more than twice as likely. Second, it is unlikely the lack of effect on latency in the Pavlovian rats was due to the slightly longer response latencies (making them more “difficult” to disrupt), as we found no relationship between port entry latency during the vehicle session and the change in latency following VP inactivation in either the instrumentally trained group (r(12)=-0.035, p = 0.91) or the Pavlovian conditioning group (r(12)=0.05, p = 0.88), suggesting that functional effects on latency in the instrumental task were not driven by those rats that had shorter latencies.

2) It appears as though the history of reward for the two tasks is different, which in principle could lead to different learned valuation of the cues or actions taken by the animals. Can the authors discuss, or control for this in their analyses?

As the reviewer points out, because the delivery of reward is contingent on the rats’ behavior in the instrumental task, but not the Pavlovian, it is possible for differences in reward history to arise both between tasks and between rats. We have aimed, with the design of the training procedures for the instrumental task, to reduce the number of cue presentations in which the animal does not receive reward by presenting longer cues at the start of training, which progressively decrease in length across training days. To determine the impact of immediate reward history during recording sessions we assessed the number of the last 10 trials during which the animal made a response and whether that affected port entry latency on the current trial, and whether this differed depending on the relationship between port entry response and reward delivery (instrumental vs Pavlovian training). We found no effect (F(1,611)=0.03, p = 0.85), regardless of training history (F(1,611)=0.25, p = 0.61). Though our tasks were not designed with sufficient trials to reliably assess the impact of reward history within session on neural encoding, to assess whether neural encoding of latency was influenced by the number of cue-reward or cue-action-reward pairings on a larger scale, such as through overtraining, we assessed the impact of session number on the correlation coefficients between neural firing and port entry latency. We found no effect (F(1,691)=2.04, p = 0.15).

**Author response image 3. respfig3:** 

3) The authors describe the results in terms of incentive and conclude as to what the VP does not do (general motor envigoration) but remain extremely vague as to what they propose the VP does do ("encode a motivational signal that may be uniquely integrated in the VP"). Even the description in terms of incentive/motivational signal is a bit problematic – how could this produce distinct effects on reaction times (effects absent) from effects on choice (port entry – effects present) in the Pavlovian. The authors should ground their results in a theoretical proposal and discuss how their results support this theory in the discussion.

We have revised this part of the Discussion section to be more explicit about the consistency between our results and current theoretical proposals, as well as the lack of explicit predictions about dissociations between response probability and response vigor or latency made by any current theoretical proposals of which we are aware. We disagree that framing VP neural responses that predict latency as an incentive or motivational signal is problematic. Our proposal, that cue-elicited activity in VP acts as an incentive signal, does not necessitate that incentive signals have distinct effects on reaction time versus response probability, as after instrumental conditioning, VP activity predicts and contributes to both. In the case of Pavlovian conditioning, the relationship between VP activity and response probability (which is weaker than after instrumental conditioning) may be driven by encoding of incentive value or may be related to a dissociable function of VP, such as encoding of reward expectancy.